# Risk Factors for Non-Communicable Diseases in Refugees, Asylum Seekers, and Subsidiary Protection Beneficiaries Resettled or Relocated in Portugal Between 2015 and 2020

**DOI:** 10.3390/ijerph21111505

**Published:** 2024-11-13

**Authors:** Ana Pinto de Oliveira, Cláudia Conceição, Inês Fronteira

**Affiliations:** 1Faculty of Medicine and Biomedical Sciences, University of Algarve, Gambelas Campus, 8005-139 Faro, Portugal; 2Global Health and Tropical Medicine, Institute of Hygiene and Tropical Medicine, NOVA University of Lisbon, 1349-008 Lisbon, Portugal; claudiaconceicao@ihmt.unl.pt; 3Public Health Research Centre, NOVA National School of Public Health, NOVA University Lisbon, 1600-407 Lisbon, Portugal; ines.fronteira@ensp.unl.pt; 4Comprehensive Health Research Center, NOVA National School of Public Health, NOVA University Lisbon, 1600-407 Lisbon, Portugal

**Keywords:** noncommunicable diseases, biological risk factors, behavioral risk factors, resettlement, refugees, asylum seekers, subsidiary protection beneficiaries

## Abstract

Non-communicable diseases, previously thought of as a problem of high-income countries, now coexist in low- and middle-income countries, including the countries of origin for many refugees traveling to Europe. We aimed to describe the prevalence of risk factors for non-communicable diseases among refugees, asylum seekers, and subsidiary protection beneficiaries resettled or relocated in Portugal between 2015 and 2020 and compare these to the prevalence of risk factors in the 12 months before they left their country of origin. A cross-sectional study was conducted between 2019 and 2020 of all refugees, asylum seekers, and subsidiary protection beneficiaries attending a Lisbon, Portugal refugee center. Behavioral and biological risk factors were assessed using the WHO STEPwise modified questionnaire. A descriptive statistical analysis was conducted, which included 80 respondents, mainly men, with an average age of of 30.3 ± 9.8 years. The prevalence of several behavioral risk factors for non-communicable diseases among refugees, asylum seekers, and subsidiary protection beneficiaries was higher at the time of the study than in the 12 months before leaving the country of origin. Differences between men and women were noted in tobacco (49.1% vs. 25.9%) and alcohol use (43.4% vs. 18.5%) in the receiving country. Overweight and obesity also showed differences by gender (7.5% vs. 11.1% and 39.6% vs. 48.1%). The prevalence of suicidal ideation and suicidalplanning was high, and varied from 6.3% and 20% in the country of origin to 16.3% and 38.5% respectively in the receiving country, however the prevalence of suicide attempts was lower in the receiving country (66.7%) compared to the country of origin (100.0%). Information on health and social determinants is critical to identify priorities and increase access to access to gender-specific health and community level interventions, including mental health, to reduce risk factors associated with refugee relocation and resettlement.

## 1. Introduction

Non-communicable diseases (NCDs) represent a significant public health and broader societal problem, causing approximately 42 million deaths worldwide every year and leading to high costs in health expenditures, absenteeism, and loss of years of productive life [1]. Current disease trends suggest a worsening situation over the next decade, with the World Health Organization (WHO) projecting 55 million deaths from NCDs annually by 2030 and an estimated cumulative loss of output of $47 trillion between 2011 and 2030 [2,3]. Therefore, it is unsurprising that several NCD-related indicators have been included in the list of Sustainable Development Goals (SDGs) adopted by the United Nations General Assembly in July 2017 [4,5].

The five subgroups of NCDs that are being targeted globally for control include cardiovascular disease (CVD), cancer, chronic respiratory disease (CRD), type 2 DM (DM2), and mental health, along with selected NCDs risk factors (RFs), such as smoking and alcohol consumption, unhealthy diet, and physical inactivity, biological RFs, such as high blood pressure (HBP), high blood glucose and high cholesterol, and overweight and obesity, which are identified as major underlying causes of NCDs [6,7,8].

NCD rates tend to be higher among communities that are disadvantaged and socially excluded, mainly in high-income countries [9]. Migrants are frequently among the most socially vulnerable populations, and they are disproportionally affected by NCDs compared to the host populations [10]. Evidence suggests that NCDs are among the most prevalent health conditions in refugees [8,11]. While acute infectious diseases remain a priority in some refugee contexts, NCDs make up a more significant share of the burden of those displaced from conflicts in Iraq, Syria, and Ukraine. These parallels a global trend in which deaths due to infectious diseases have declined, life expectancy has increased, unhealthy lifestyle behaviors have spread to developing economies, and the overall proportion of deaths due to NCDs has risen to more than 50%.

NCDs and an inability to address related behavioral RFs tend to result in poor outcomes that increase morbidity and mortality in migrants and refugees. Although often seen as chronic and not an immediate priority, not prioritizing NCD, even in an emergency setting, can have significant adverse impacts on long-term health outcomes for refugee and migrant populations. For those suffering from CVD, DM2, CRD, HBP, and cancer, the long, arduous journeys to and through Europe frequently exacerbate these conditions and interrupt the continuous treatment essential for their management [12]. Furthermore, access to quality health and care services in transition and/or destination countries is underpinned by availability, accessibility, accommodation, affordability, and acceptability, as well as by the ability of refugee and migrant users to understand, use, and engage with the health and care services [13,14]. The dissonance between these populations and transit or host healthcare services is manifested through multiple structural, physical, financial, sociocultural, and communicative barriers [15,16,17,18]. A lack of health literacy compromises their efficacy in navigating health systems. It is exacerbated by a lack of legal status, especially when inclusive health policies and social-economic protection (e.g., social safety nets) are absent or frail [19,20].

The WHO has recommended the surveillance of common RFs for NCDs using the STEPwise approach to surveillance (STEPS) to monitor trends within and across countries [21]. The organization states the global fight against NCDs requires data disaggregated by migratory status. Studies have shown that refugees and migrants have lower prevalence rates of NCDs on arrival in destination countries than in the host population, but that rates begin to converge as the duration of stay in the host country increases. The STEPS on NCDs among the Syrian refugee community in Turkey showed that 58.7% of the refugees were at high risk of developing NCD [4].

This study aimed to describe the behavioral and biological RFs for NCDs in refugees, asylum seekers, and subsidiary protection beneficiaries resettled or relocated in Portugal between 2015 and 2020 and compare the prevalence of these RFs in the 12 months before they left their country of origin. Based on past evidence, we hypothesized that the prevalence of RFs for NCDs is higher in the receiving country. The conditions surrounding the migration and resettlement/relocation process may increase exposure and vulnerability to RFs for NCDs.

## 2. Materials and Methods

### 2.1. Study Design and Context

A cross-sectional descriptive study occurred between September 2019 and December 2020 at the Refugee Temporary Shelter Center (in Portuguese, CATR) in Lisbon, Portugal. The questionnaire survey was carried out by a principal investigator who interviewed and physically examined the participants. An interpreter was present whenever the participants could not speak English or Portuguese.

Two thousand one hundred eighty-one refugees, asylum seekers, and subsidiary protection beneficiaries arrived in Portugal between 2015 and 2020. Of these, the Jesuit Refugee Service (JRS) received 343 at the CATR, of which 80, because they lived in the Lisbon metropolitan region and were aged 18 years or more, were invited to participate in this study. We used a modified form of the modular expanded STEPS that incorporates time (i.e., 12 months before leaving the country of origin and at the time of application of the questionnaire) into almost every question (Appendix A). A pre-test of the modified instrument was conducted among refugees and asylum seekers received and integrated by organizations other than JRS. The following data were collected: sociodemographic information, behavioral, and biological RFs for NCDs, according to the STEPS Manual [22]. Arterial blood pressure was measured using an automated standard digital sphygmomanometer (Tensiometer Digital Pic Cs 530 Wrist), and weight was measured in kg using a person scale (Rowenta BR2520V0).

### 2.2. Data Analysis

All the statistical analyses were performed with the Statistical Package for the Social Sciences (SPSS) v24 software. Continuous variables are presented as the means and standard deviations (±SDs), and median, minimum, maximum, and categorical variables are presented as counts and relative frequencies.

### 2.3. Research Ethics

Ethical approval was granted by the Regional Health Administration of Lisbon and Tagus Valley Ethics Committee (Approval 022/CES/INV/2019). Written informed consent was obtained before participants were enrolled in the study using the STEPS survey consent form. We followed STROBE guidelines for reporting observational studies (S1 STROBE Checklist) [23].

## 3. Results

In the present study, 80 refugees, asylum seekers, and subsidiary protection beneficiaries were invited and agreed to participate (100% response rate).

### 3.1. Sociodemographic Characteristics

The refugees, asylum seekers, and subsidiary protection beneficiaries were mainly men (66.3%), aged between 18 and 29 years (53.0%), had never been married (55.0%), had been born in the Eastern Mediterranean WHO region (56.3%), applied to the relocation program in Greece, Malta and/or Italy (56.3%), and waited between asylum application and reception in Portugal for approximately 35.8 months in rented rooms/houses (40.0%) or refugee camps (36.3%). In addition, 66.3% had no education or attended the equivalent of elementary school (Table 1).

At the time of the study, 77.5% of the participants were unemployed and had an average monthly income of 267.5 euros. The primary work sector for those working was the tertiary sector—nongovernmental Organizations (NGO). In the 12 months before leaving the country of origin, the majority were employed (86.2%), mainly in NGOs (31.3%), with an average monthly income of 220.3 euros (Table 2).

### 3.2. Behavioral Risk Factors for NCDs

Before leaving the country of origin, the consumption of smoked tobacco by men and women participants was similar, but at the time of the study, men were more likely to be smokers (49.1%) than women (25.9%). The daily consumption of smoked tobacco was lower at the time of the study (87.9%) than at 12 months before leaving the country of origin (100.0%). Approximately 13.0% of smokers had been advised by a doctor or other health professional to quit smoking in the 12 months before leaving their home country, while 21.0% had quit smoking at the time of the study. In the 12 months before leaving their country of origin, the percentage of participants exposed daily to passive smoke at home was lower than that at the study time (30.0% vs. 50.0%). The same trend was observed for environmental exposure at the workplace (33.3% vs. 66.6%) (Table 3).

Most participants reported not consuming alcoholic beverages in the 12 months before leaving their country of origin (68.1%). At the time of the questionnaire, only 38.1% did not consume alcoholic beverages.

The lifetime prevalence of consuming at least one alcoholic beverage was greater among men participants (20.8% vs. 43.4%) than among women participants (11.1% vs. 18.5%). Among consumers, in the 12 months before leaving the country of origin, alcoholic beverages were consumed most frequently monthly (28.6%) and weekly (42.8%) at the time of the study (Table 3).

The prevalence of binge drinking was more significant at the time of the questionnaire (28.5%) than at 12 months before leaving the country of origin (17.9%). However, the average number of drinks consumed was more significant in the 12 months before leaving the country of origin (12.1 vs. 7.0 drinks) (Table 3).

The proportions of participants who did not meet the WHO recommendation on fruit and vegetable consumption in the 12 months before leaving the country of origin and at the time of the questionnaire were 2.5% and 31.3%, respectively. Approximately 16.3% of participants always or often ate processed food with a high salt content in the 12 months before leaving the country of origin (vs. 22.6%) (Table 3).

Most participants did not engage in physical activity; 22.5% had a moderate intensity of physical activity of less than 150 min per week in the 12 months before leaving the country of origin, while 38.8% had a moderate intensity of physical activity at the time of the questionnaire (Table 3).

In the 12 months before leaving the home country, 33.8% of the study population consulted a health professional and reported that they were counseled about reducing smoking (100.0%), alcohol consumption (77.8%), salt (81.5%) and fat intake (85.2%), increasing the consumption of fruits and/or vegetables (96.3%), physical exercise (81.5%), and maintaining an ideal weight (77.8%). However, consultation and lifestyle counseling with a health professional in the present study were less common (23.8%) at the time of the questionnaire, as was the type of risk factor reduction given (Table 3).

Most women participants reported not having cervical-vaginal cytology performed in the 12 months before leaving their country of origin (92.6%) or at the time of the questionnaire (85.2%). Of the nine women aged 30–49 years, three had cervical-vaginal cytology performed, two in the 12 months before leaving the country of origin, and one at the time of the questionnaire (Table 3).

The prevalence of self-reported HBP was 16.3% at the time of the questionnaire. Since their arrival in the receiving country, 21.2% of participants had never measured their blood pressure, and more than one-third (37.5%) had not measured it in the 12 months before leaving the country of origin. Of the total individuals with HBP, 75.0% reported taking antihypertensive medication during the 12 months before leaving the country of origin, while 38.5% reported taking antihypertensive medication at the time of the questionnaire.

Suicide attempts in 100% of participants who planned suicide were recorded in the 12 months before leaving the country of origin, and suicide attempts were recorded in 66.7% of participants who planned suicide at the current time. Seeking health professionals for suicidal ideation was reported in 20.0% and 38.5% of participants 12 months before leaving the country of origin and at the time of the questionnaire, respectively (Table 3).

### 3.3. Biological Risk Factors for NCDs

The overall prevalence of pre-HBP or HBP, measured on site, was 56.3%, with men participants presenting a higher prevalence (69.8%) than women (29.6%). According to the BMI, 42.5% of the participants were overweight, and 18.6% were obese. A total of 21% of the population had increased, and 27.5% had significantly increased abdominal circumference. The percentage of women with an increase or substantial increase in abdominal circumference (29.6% vs. 59.2%, respectively) was much more significant than that of men (16.9% vs. 11.3%). Thus, BMI and central adiposity (abdominal circumference) were more significant among women, increasing the risk of NCDs (Table 4).

### 3.4. Combined Risk Factors for NCDs

The following RFs were identified in participants at the time of the questionnaire: daily tobacco use, having fewer than five servings of fruits and/or vegetables per day, having less than 150 min of moderate activity per week or equivalent, overweight or obese, and HBP (Table 5). The prevalence of the combined RFs for NCDs ranged between 5.0% (four RFs) and 28.8% (two RFs), with 24.0% of the participants not having any RFs (Table 5).

## 4. Discussion

This study aimed to describe the behavioral and biological RFs for NCDs in refugees, asylum seekers, and subsidiary protection beneficiaries who were relocated or resettled in Portugal between 2015 and 2020 by comparing the 12 months before leaving their country of origin with the time of the questionnaire. The main results of this study regarding the participants’ behavior showed that RFs for NCDs were higher in the country of origin and more prevalent among male participants who have two or more of the five combined RFs. Although participants’ access to healthcare was higher in the receiving country compared to the country of origin, screening, and counseling on lifestyles with an impact on the incidence of NCDs was lower in the receiving country. At the time of the study, biological RFs were prevalent among participants, with more than half of the participants having hypertensive or a pre-hypertensive condition.

The NCDs determinants behind health disparity among “refugees” are well documented and include the adoption of unhealthy behaviors related to the new sociocultural contexts such as poorer socioeconomic conditions, e.g., vulnerable living and working conditions, reduced access to information and services. Refugees often face difficulties in accessing and using health services despite the Portuguese National Health System (NHS) being universal for all residents without discriminating against their socioeconomic status, professional situation, or legal status. The State Budget finances the system and does not charge citizens, allowing tendentially free access. Thus, the Portugal Country Health Profile 2019 report, State of Health in the European Union states, “All immigrants who are in Portugal for over 90 days have access to GP services, irrespective of their legal status. There are no restrictions for pregnant women, children, people with infectious diseases, or those needing urgent care. While the NHS covers all residents in Portugal, there are barriers that, indeed, prevent immigrants from accessing NHS services, including language and cultural differences” [24] (p. 16). One of the contradictory aspects is that although policies protect these citizens, barriers prevent access to medical care. These include economic and legal constraints, lack of information about health rights and other individual, sociocultural, economic, and administrative reasons. As a result, they often tend to be missed by conventional disease prevention and surveillance programs.

Even though there are biological predispositions to NCDs, it is estimated that if the leading social and behavioral causes were eliminated, 80% of CVD and DM2, as well as 40% of cancers, could be prevented [25]. Refugee health is strongly related to the social determinants of health, such as lifestyle choices, social and community impact, living/working circumstances, and broad socioeconomic, cultural, and environmental factors [25,26]. This study’s results will be analyzed by considering the social determinants of health and lifestyle choices.

### 4.1. Sociodemographic Characteristics

Sociodemographic characteristics such as age, gender, education level, and employment status are known to play a decisive role in the decision to migrate, and they are also determinants of risk factors for NCDs. Participants were, on average, 30 years old, mostly men, single, unemployed, with no education or with the equivalent of primary education, and from the Eastern Mediterranean region.

In resource-constrained situations, their families may prioritize young men and adolescent boys for migration; it may be less acceptable for women to move or travel in some contexts. These age and gender patterns are broadly consistent across some of the largest refugee groups, including those from Syria, Iraq, Iran, and Somalia [27]. The large proportion of young men compared to the number of young adult women suggests that many refugees from these countries are young men traveling alone.

This study used the education level to indicate an individual’s socioeconomic position. With most participants, 66.3% (n = 53), having only attained primary education, it can be expected that the population represented would possess more risky behaviors, thus negatively exposing them to the development of NCDs. This is in line with reports by Nawiet et al. (2006), who conducted an Indonesian study, and showed that socioeconomic position is inversely proportional to the average number of RFs for NCDs, and populations with low socioeconomic status, as determined by both income and level of education, are therefore relatively at greater risk of developing NCDs [28].

The findings of our study also showed that most of the refugees, asylum seekers, and subsidiary protection beneficiaries live in poverty and maintain their lives with low income (mean mensal income of 265.7 euros) and high unemployment levels (77.5% of participants) at the time of the study. These conditions are conducive to the lack of control over various factors influencing health and limits refugees’ opportunities to make healthy choices in life. The resulting consequences include poor cognitive functioning, lower educational achievement, increased risks of low income, unstable job status, smoking, alcohol abuse, physical inactivity, and an unhealthy diet [29], factors associated with NCDs. These factors significantly influence health by determining material circumstances, housing types, and locations [30]. An example would be the link between socioeconomic status and NCDs as manifested directly and indirectly through the effects on access to healthcare facilities and preventative medical examinations [31]. The restrictions to access and utilization of prevention and care services for the poor communities are, for example, related to an inability to afford transport to healthcare facilities.

The sociodemographic characteristics of the study population, as observed, seem to make them vulnerable to the main known RFs for NCDs, given their male gender, low level of education, and low socioeconomic status.

### 4.2. Behavioral Risk Factors for NCDs

NCDs result from an accumulation of risk factors that derive from social determinants of health, behavioral risk factors, and genetic predispositions. The stress of migration and adaptation to new cultures can promote alcohol, tobacco use, and substance abuse. Evidence shows that a significantly more significant percentage of men in most age groups have risk behaviors [27]. This includes smoking, alcohol consumption, insufficient intake of fruits and vegetables, increased salt intake, and frequent consumption of processed foods. On the other hand, a more significant percentage of women in older age groups have the most biological RFs, such as overweight and obesity, HBP, glucose, and cholesterol levels [27,31]. Our findings have revealed a consistent pattern of gender inequalities disfavoring men in reporting behavioral RFs for NCDs. Economic and social factors can partly explain this. The impact of professional factors can be explained by the fact that refugee and migrant women are less frequently employed than men. When employed, they work in less regulated sectors, where they experience greater precariousness and have less access to social security [32]. Marmot et al. (2012) state that participation in the labor market determines a wide range of life opportunities, mainly through regular wages and social status [33]. Threats to social status and identity due to instability or job loss can negatively affect health and well-being, similar to economic stressors.

#### 4.2.1. Tobacco and Alcohol Consumption

Tobacco consumption contributes to the majority of NCDs, including CRD, cancer, CVD, and DM2. The literature has attributed health risk behaviors, such as smoking, to the adoption of Western lifestyles among populations in low-income countries [34]. In this study, smoking and alcohol use were found to be more prevalent at 12 months before leaving the country of origin. The prevalence of tobacco consumption in the 12 months before leaving the country of origin was identical to that described for the WHO African (18.5% in 2020) and Eastern Mediterranean (18.6% in 2020) regions [34]. In contrast, the prevalence of tobacco consumption at the time of the study was almost twice as high as that reported for the native Portuguese population and six times greater than that reported for daily consumption [35,36]. The available literature regarding tobacco use among refugees in host countries is relatively sparse. Only one study assessed tobacco use and showed that Vietnamese refugee groups had a greater prevalence of tobacco use than native populations. In comparison, Ukrainian refugees had a lower prevalence of tobacco use [37]. This study focused only on new arrivals and is not a true reflection of tobacco use after refugee resettlement over time. Tobacco consumption remains prevalent in refugee countries of origin around the world. Still, behaviors related to tobacco and other substance use by refugees in the host country are not described in the literature [38].

Although alcohol consumption is elevated in high-income countries in Europe and America, the lowest alcohol consumption is observed in most regions of North Africa and the Eastern Mediterranean (0–20%); in the Sub-Saharan African region, the prevalence of consumption ranges between 20 and 40%, while in the European region stands at 21% [39]. In 2019, Portugal recorded regular alcohol consumption in 21% of the population aged 15 and older [38]. The results of the present study support these findings, with a greater prevalence of alcohol consumption among study participants in Portugal than among those in the 12 months before leaving the country of origin in both genders. Also, the results are in line with the literature, which suggests that lower socioeconomic groups suffer a higher incidence of alcohol-related non-communicable diseases [40]. The literature on refugees and binge drinking is sparse [41]. Most studies have focused on various settings, for example, in Australia, Uganda, Georgia, and South Korea, but comparative studies are rare [42,43,44,45,46,47,48,49,50,51,52,53,54,55]. Other cross-national studies compare harmful substance use among displaced populations but do not distinguish between alcohol and other drugs or harmful substance use on general health status or mental health risks associated with migration [54,55,56,57,58,59,60]. The vast and different national contexts in terms of conflict, legal and policy frameworks, adhesion to international norms and treaties, and local norms and treaties and support systems make international comparisons difficult. In addition, some studies show that the cultural and ethnic backgrounds of different displaced populations contribute to shaping their vulnerability to alcohol consumption. However, there are some common denominators, particularly regarding RFs for the development of harmful drinking. As several authors note, men tend to drink more than women, which may reflect specific cultural attitudes toward gender-appropriate behavior [43,44,46,50,52,54,60,61,62,63,64].

Another commonly cited RF is trauma (pre- or post-displacement), especially in the context of forced displacement because of armed conflict, the loss of associated property, social relationships, livelihoods, and identity [46,65,66,67]. It has been suggested that harmful substance use may act as a coping strategy in response to these stressors and that alcohol consumption may constitute an attempt at behavioral self-medication [67,68,69,70,71,72].

#### 4.2.2. Dietary Practices

A poor diet is one of the leading preventable causes of NCDs, including obesity, cancer, CVD, and DM2. It is characterized by low fruit and vegetable consumption and high salt, fat, and sugar intake. This study reported substantially reduced consumption of fruits and vegetables in the host country compared to previous consumption in the home country and increased consumption of processed foods in the host country. The difference in fruit and/or vegetable consumption observed may be attributed to the socioeconomic situation—low income—of refugees settling in high-income countries, a key determinant of food insecurity [73,74].

Additionally, upon arrival in the host country, refugees may retain, adapt, or exclude traditional foods and adopt food patterns/choices, in part or whole, through “dietary acculturation” [74]. Some studies show that after migrating to the host country, migrants may have difficulty familiarizing themselves with the new food environment, which may result in food insecurity because it depends on economic constraints, time, literacy in the host country language, and social support, among other factors [74].

Several studies have shown that refugees and migrants from various countries are indeed challenged by facing a foreign food culture, which often leads to unhealthy food choices or a so-called Westernization of eating habits, e.g., consumption of prepared and fast food, which is high in calories and low in nutritional value [75,76]. The low income and limited resources at the individual, household, and community levels, associated personal powerlessness, and lack of choice combine to predetermine and constrain individual options for diet, nutrition, and activity [77].

#### 4.2.3. Physical Activity

Physical inactivity or sedentary lifestyles have become a significant public health concern, contributing significantly to global NCD morbidity and mortality rates, which have been primarily attributed to urbanization and global socioeconomic transitions [78]. This study revealed inadequate physical activity levels in the host country, higher than those recorded 12 months before leaving the country of origin. This difference may result from inequalities at the individual, group, or environmental levels regarding access to the conditions necessary for physical activity or the inability to communicate in the local language, reducing refugees’ access to information related to physical activity, including exercise techniques and locations [79,80].

Financial constraints are also common immigration stressors that limit refugees’ ability to pay for gym memberships or to purchase sports apparel/equipment [81]. Physical activity has a well-documented positive impact on health; it can prevent or delay the onset of a variety of mental disorders and may also have therapeutic benefits when used as a stand-alone or adjunctive treatment [79,80,81,82]. Some studies have reported that the benefits of physical activity are well-known among refugees. However, the most significant barrier to their participation is a lack of familiarity and comfort in taking the first steps toward physical activity in the host country [82].

Most refugees also stated that they had been less physically active since their arrival in the host country than before they fled their home countries; it is plausible that newly arrived refugees felt they had more important priorities than physical activity [80]. Furthermore, a study among Somali refugee women in New Zealand revealed that physical activity improved refugees’ social development and community cohesion in their new home country [83]. Other studies have confirmed these findings [84,85]. Therefore, physical activities can be an effective way for refugees to improve their mental health, helping them cope with circumstances in their host countries and their memories of their past.

#### 4.2.4. Access to Healthcare

Routine consultations, laboratory tests, and screenings are used for the early diagnosis of some commonly occurring health problems because of early management and treatment to reduce complications and the risk of death [86]. The differences observed regarding preventive healthcare in the 12 months before leaving the country of origin and at the time of the study reflect differences in the frequency of other risk factors, such as higher alcohol, tobacco, and salt consumption and physical inactivity. Our results revealed greater use of preventive healthcare by refugees at the time of the study than at 12 months before leaving the country of origin but lower health literacy in the dimensions of disease prevention and health promotion. The observed differences may result from language issues, cultural or economic factors, and the demands placed on health professionals encountering refugees in different situations. The cervical cancer screening adherence rate among refugee women in our study was lower than that among those 12 months before leaving the country of origin and among native Portuguese women (88.8% in 2020) [87]. This difference may be explained by the inexistence or inaccessibility of cancer prevention services in the countries of origin, psychosocial (shame and embarrassment), or communication barriers (inability to speak Portuguese) [88,89].

Several strategies can enhance access to primary healthcare for refugees, asylum seekers, and beneficiaries of subsidiary protection. These include fostering multidisciplinary collaboration, providing free or low-cost services, offering neighborhood-based services, and arranging free transportation to and from appointments. Additionally, extending the operating hours of healthcare facilities and ensuring that service providers are sensitive to gender needs can further improve access.

To assist refugees, asylum seekers, and beneficiaries of subsidiary protection in navigating healthcare systems, it is essential to support them in registering, booking appointments, and attending services. This could involve hiring interpreters to provide clear explanations of unfamiliar clinical processes and treatments, ensuring timely management of healthcare needs.

While non-governmental organizations (NGOs) play a role in complementing healthcare provision in some contexts, their ability to deliver continuity of care and refer individuals to specialized services is often not sustainable.

#### 4.2.5. Mental Health

In the refugee population, mental health mortality is low and is almost exclusively related to suicide [90]. Suicide, in turn, occurs mainly in people with severe mental illnesses, most of which are treatable (major depression and bipolar disorder) and belong to the group of potentially preventable deaths, provided that the diagnosis of the underlying pathology is made promptly and the therapeutic approach is effective [91]. We observed a greater prevalence of suicidal ideation at the time of the study than at 12 months before leaving the country of origin. A combination of socioeconomic disadvantage, exposure to potentially traumatic events, increased depression and anxiety, or a lack of adequate and accessible healthcare may account for this disparity [90]. Refugees and asylum seekers often face delayed mental health diagnoses, treatment, and care that can lead to poor health outcomes in the long term. Professional help-seeking for suicidal ideation was greater in the host country, which may explain the lower number of suicide attempts in the host country than in the country of origin. This is demonstrated by the alternative strategies for accessing health services, including mobilizing the institution’s resources, such as the JRS Mental Health Office. This alternative response has overcome some of the shortcomings of the Portuguese National Health Service. Although Law No. 26/2014 of 5 May (Article 73) stipulates that ‘refugees’ fall into particularly vulnerable groups of people who are guaranteed adequate healthcare, including treatment for mental disorders, when necessary, under the same conditions as nationals, there are barriers that affect the issue of access to healthcare.

The perceived positive experience with the healthcare provided by the JRS (Mental Health Office) may cause refugees to seek care when suicidal ideation or attempts are made, which may be reflected in our results.

#### 4.2.6. Biological Risk Factors for NCDs

At the time of the study, more than half of the participants had HBP or pre-HBP. This finding deserves special attention because it occurs in a population with a mean age of 30, demonstrating the increased incidence of this health problem in younger age groups. The review by Rada and Cabieses (2024) indicated that international migrants face various challenges in their migration process, limiting their access to preventive services. The literature cites the impact of the structure and functioning of health systems at origin, transit, and destination, in addition to the complex and multidimensional interaction of knowledge of health rights, the health system and services available, migratory status, immigration and health policies, as well as financial, cultural and linguistic barriers, as the most significant determinants of HBP prevention and control in migrants [92].

Overweight affects 42.5% of the study’s participants, who consequently had a substantially increased risk of metabolic complications. Obesity and overweight are mainly represented as the result of poor individual eating patterns, ill-informed decisions about diet and nutrition, and exercise (or the lack of it). The low income can explain the results of our study as a key factor in restricting refugees’ food choices and physical activity levels. After resettlement in the host country, food insecurity is prevalent among refugee families. The experience of economic hardship and lack of social support distinguishes many refugees from other immigrants, making them more vulnerable and at higher risk of developing unhealthy behaviors because of enduring these situations [93].

One qualitative study found that limited finances were a barrier when it came to healthy eating and concluded that the socioeconomic challenges faced by refugees after resettlement were significant factors in negatively altering their eating patterns [94]. A second qualitative study supported this view, revealing how limited finances were a barrier to accessing sports facilities and subsequently affected physical activity levels among refugee adolescents [80]. The high cost of healthy food was also an element that shaped diet [95].

### 4.3. Limitations of the Study

This study has several limitations. The first limitation is the small number of study participants. This restriction arises because the subjects were not sampled from the entire population but rather from a specific subpopulation selected based on certain criteria, specifically those who were welcomed and integrated into the CATR between 2015 and 2020. However, the study results highlight a problem that should be addressed in public policies in Portugal, even though there is insufficient data to conclude about the entire population.

Another limitation is that participants were asked about events that occurred in the past, so it is assumed that memory bias may exist [96]. The data were collected through face-to-face interviews with interpreters (“interpreter effect”), which can influence interviews (“interpreter effect”) [97]. Interpreters can affect response behavior due to their presence, behavior, and external characteristics (e.g., sex and age), thus creating bias.

Another limitation could be the possible non-differential or differential underestimation by legal status (refugee, asylum seeker, and beneficiary of subsidiary protection). However, we would have no reason to believe that the effect of legal status on the prevalence of risk factors for NCDs would differ between refugees, asylum seekers, and subsidiary protection beneficiaries since, in Portugal, everyone has equal access to healthcare.

## 5. Conclusions

This study, using the WHO’s STEPwise approach, revealed that refugees, asylum seekers, and subsidiary protection beneficiaries have more RFs for NCDs than they had 12 months before leaving their country of origin, suggesting that the ‘refugee’ population faces living conditions that can affect their state of health. The WHO STEPwise approach to NCD risk factor surveillance highlights the importance of collecting data on biological RFs, including overweight and obesity, and on tobacco use, alcohol use, physical inactivity, and unhealthy diet. In this study, as in other guidelines to address the causes of and reduce the incidence of NCDs, little attention is given to tracking the social and cultural determinants of health. The prevention and management of NCDs need to address the root causes, which are predominantly based on social and environmental determinants [98].

Although Portugal has a universal health system and regulations that allow refugees to access health services, considerable inequalities persist and are more evident among migrant populations, ethnic minorities, and other vulnerable groups due to system weaknesses that include limitations linked to political, socioeconomic, community, organizational, and personal aspects.

Nevertheless, the results of this study reinforce that it is essential to have a greater knowledge of the “refugee” population in each context, which includes their health and social determinants, identifying priorities for intervention, assessing specific needs, and establishing integrative and sustained health policies and strategies that produce real effects in reducing risks and vulnerabilities and allowing for effective health gains.

## Figures and Tables

**Table 1 ijerph-21-01505-t001:** Distribution of the population under study according to sociodemographic characteristics.

Demographic Information		n	%
Sex			
Men		53	66.3
Women		27	33.7
Age Group (in years)			
18–29		44	53.0
20–29		1	1.3
30–44		29	36.3
45–59		5	6.3
60–69		1	1.3
Marital Status			
Unmarried		49	61.2
Currently married		31	38.8
Level of education			
No formal schooling		12	15.0
Less than primary school		2	2.5
Primary school complete		39	48.8
Secondary school completed		0	0.0
High school completed		14	17.5
College/University completed		13	16.3
Postgraduate degree completed		0	0.0
Country of origin (WHO region)			
African	Sub-Saharan	34	42.5
	Mediterranean	1	1.2
Eastern Mediterranean		45	56.3
Asylum application country			
Greece/Italy/Malta		45	56.3
Turkey/Egypt		30	37.5
Portugal		5	6.3
Place of residence between asylum application and relocation/resettlement			
Refugee camp		29	36.3
Refugee center		8	10.0
Rented house/apartment		32	40.0
Rented room		5	6.3
Other		6	7.5

**Table 2 ijerph-21-01505-t002:** Distribution of the population under study according to employment situation and mean monthly per capita income.

	12 Months Before Leaving the Country of Origin		Time of the Questionnaire	
	n	%	n	%
Employment status				
Employed	69	86.2	18	22.5
Unemployed	11	13.8	62	77.5
Mean mensal per capita income				
Mean ± SD.	220.3 ± 258.1		267.5 ± 265.7	
(Minimum maximum)	(0–1000)		(150–1500)	

**Table 3 ijerph-21-01505-t003:** Distribution of the study population according to behavioral RFs for NCD.

			12 Months Before Leaving the Origin Country		Time of the Questionnaire	
			n	%	n	%
Behavioral Risk Factors						
Tobacco use						
Tobacco use		Men	10	18.9	26	49.1
		Women	5	18.5	7	25.9
Smoking status (^a^ n = 15; ^b^ n = 33)		Daily	15 ^a^	100.0	29 ^b^	87.9
		Not daily	0	0.0	4 ^b^	12.1
Second-hand smoke exposure						
At home			24	30.0	40	50.0
At Workplace (^c^ n = 69; ^d^ n = 18)			23 ^c^	33.3	12 ^d^	66.6
Alcohol use						
Alcohol use	Men		11	20.8	23	43.4
	Women		3	11.1	5	18.5
Heavy episodic drinking			4	17.9	5	28.5
Dietary Practices						
Inadequate consumption of fruits and/or vegetables			2	2.5	25	31.3
Frequent or often consumption of processed foods			13	16.3	18	22.6
Physical activity						
Insufficient physical activity			18	22.5	31	38.8
Access to healthcare						
Consultation with healthcare professional			27	33.8	19	23.8
Smoking cessation			27	100.0	11	57.9
Reduce alcohol use			21	77.8	2	10.5
Consumption of fruits/vegetables			26	96.3	4	21.1
Reduce salt consumption			22	81.5	5	26.3
Reduce fat in the diet			23	85.2	4	21.1
Physical exercise practice			22	81.5	3	15.8
Maintain optimal weight			21	77.8	2	10.5
Cervical cancer screening (n = 27)			4 ^e^	14.8	2 ^e^	7.4
Raised blood pressure self-reported			4	5.0	13	16.3
Mental Health						
Suicidal ideation			5	6.3	13	16.3
Search for health professionals by suicidal ideation (^f^ n = 5; ^g^ n = 13)			1 ^f^	20.0	5 ^g^	38.5
Suicidal planning (^h^ n = 5; ^i^ n = 13)			3 ^h^	60.0	9 ^i^	69.2
Suicidal attempt (^j^ n = 3; ^k^ n = 9)			3 ^j^	100.0	6 ^k^	66.7

^a–k^ Number of participants questioned according to the previous question.

**Table 4 ijerph-21-01505-t004:** Distribution of the study population according to biological RFs for NCD.

			Time of the Questionnaire	
			n	%
Biological Risk Factors				
HBP or pre-HBP		Men	37	69.8
		Women	8	29.6
Overweight (BMI ≥ 25.0 kg/m^2^)		Men	21	39.6
		Women	13	41.1
Obesity (BMI ≥ 30.0 kg/m^2^)		Men	4	7.5
		Women	3	11.1
Abdominal obesity	Men	Increase (≥94 cm)	9	16.9
		Substantial increase (≥102 cm)	6	11.3
	Women	Increase (≥80 cm)	8	29.6
		Substantial increase (≥88 cm)	16	59.2

**Table 5 ijerph-21-01505-t005:** Distribution of the study population according to the number of RFs for NCDs stratified by sex.

	Men (n = 53)		Women (n = 27)	
	n	%	n	%
Combined Risk Factors				
0	9	16.9	7	25.9
1–2	30	56.5	14	51.8
3–5	14	26.4	6	22.2

## Data Availability

The data generated during and/or analyzed during the current study are available from the corresponding author upon reasonable request.

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
