# Peer review of "Risk Factors for Non-Communicable Diseases in Refugees, Asylum Seekers, and Subsidiary Protection Beneficiaries Resettled or Relocated in Portugal Between 2015 and 2020"

_ijerph, 2024, doi:10.3390/ijerph21111505_

Round 1
Reviewer 1 Report
Comments and Suggestions for Authors
This study addresses a highly relevant and understudied issue, this is the health of refugees, asylum seekers and resettled people. In addition, these populations are highly vulnerable because they are affected by many social determinants of health that have an impact on increasing the number of years of life lost in good health. However, there are several issues that are unclear or need to be improved in the manuscript:
- There are discrepancies between the objective of the abstract and the one at the end of the Introduction, and finally with that reflected in the title. This needs to be clarified. It should be noted that to estimate prevalence, appropriate denominators should be available, making it difficult to meet this objective.
- The study population should be clarified because in some places only refugees are listed and not asylum seekers and beneficiaries of subsidiary protection.
- Indicate if any validation of the modified version of STEPS has been carried out and include the literature reference of the tool.
- According to the supplementary material, it is sex that has been measured, not gender, so in table 1, gender should be replaced by sex.
- Table 2, table 3 and table 4 all reflect two points in time, so it should be checked whether there has been a significant change in the variables measured, either through the calculation of confidence intervals or through non-parametric hypothesis testing, which is appropriate for small sample sizes.
- In table 3, for smoking status, a letter ‘d’ appears for the n-value which should be a ‘b’.
- As the study was conducted in a specific population and without sampling, limitations of the study should include possible selection bias and its impact on the generalisability of the results.
- Also in the Discussion, it would be advisable to introduce the WHO framework of social determinants of health. It is important to put behavioural and biological risk factors into context, as these are influenced by structural determinants.
- In general, for the text of the manuscript, in relation to the countries reached by the study population, it is more appropriate to use ‘receiving’ rather than ‘host’. In addition, it is more appropriate to speak of women and men, rather than female and male.
Author Response
Comment 1: There are discrepancies between the objective of the abstract and the one at the end of the Introduction, and finally with that reflected in the title. This needs to be clarified. It should be noted that to estimate prevalence, appropriate denominators should be available, making it difficult to meet this objective.
Response 1: Thank you! We have rephrased the objective in line with what is reflected in the abstract.
Comment 2: The study population should be clarified because in some places only refugees are listed and not asylum seekers and beneficiaries of subsidiary protection.
Response 2: Thank you! We have addressed this issue.
Comment 3: Indicate if any validation of the modified version of STEPS has been carried out and include the literature reference of the tool.
Response 3: The STEPS instrument was developed by WHO and covers the topics and aspects relevant to our research problem and objectives. The tool has been used since 2002 for epidemiological surveillance of NCDs and RF, resulting in several reports and studies at national, sub-national and refugee population levels. In our research we used the English version of questionnaire available at
https://www.who.int/teams/noncommunicable-diseases/surveillance/systems-tools/steps
The modification of the instrument did not change the questions asked. The period prior to leaving the country of origin was added to the questions, which refer to the current situation.
Nevertheless, a pre-test was carried out with refugees received and integrated by other organization than JRS to ascertain any questions that seem ambiguous to participants and to test their understanding and interpretation of each question.
Line 105 “A pre-test of the modified instrument was conducted among refugees received and integrated by other organization than JRS”
Comment 5: According to the supplementary material, it is sex that has been measured, not gender, so in table 1, gender should be replaced by sex.
Response 5: Thank you! We have addressed this issue.
Comment 6: Table 2, table 3 and table 4 all reflect two points in time, so it should be checked whether there has been a significant change in the variables measured, either through the calculation of confidence intervals or through non-parametric hypothesis testing, which is appropriate for small sample sizes.
Response 6: Thank you. The authors chose not to apply statistical inference tests because the study includes a small number of subjects, which is not representative enough to generalize the results. On the other hand, not using a set of randomly selected subjects also prevents the application of probability theory with the application of inference techniques.
Comment 7: In table 3, for smoking status, a letter ‘d’ appears for the n-value which should be a ‘b’.
Response 7: Thank you! We have addressed this issue.
Comment 8: As the study was conducted in a specific population and without sampling, limitations of the study should include possible selection bias and its impact on the generalisability of the results.
Response 8: Thank you! We have addressed this issue. Line 420 “The first limitation is the small number of study participants. This restriction is due to subjects that are not sampled from the whole population, but from a criterion-determined subpopulation, including having been welcomed and integrated into the CATR, between 2015 and 2020. Nevertheless, we assume that we don't have enough data to draw conclusions about entire populations.”
Comment 9: Also in the Discussion, it would be advisable to introduce the WHO framework of social determinants of health. It is important to put behavioural and biological risk factors into context, as these are influenced by structural determinants.
Response 9: Thank you! We have addressed this issue. Paragraph Line 256 to 262 and 275-280.
Comment 10: In general, for the text of the manuscript, in relation to the countries reached by the study population, it is more appropriate to use ‘receiving’ rather than ‘host’. In addition, it is more appropriate to speak of women and men, rather than female and male.
Response 10: Thank you! We have addressed these issues.
Reviewer 2 Report
Comments and Suggestions for Authors
This cross-sectional survey among 80 refugees raises and interesting research question on prevalence of risk factors and NCD's before and after arrival to the hosting country. The manuscript reads well, but there are a few major concerns:
1. The number of study participants is low and impairs making strong conclusions. Were there any sample size calculations or power considerations prior to conducting the study? Would have been worthwhile to consider what it would be possible to conclude with this study size
2. All findings depend on what the study participants report on their risk factors 12 months earlier. The authors acknowledge that this implies a huge risk of information bias, but should further consider that a number of other biases may apply as well as confounding, and this should be discussed in much more detail, preferably including the use of DAG's
3. There is no STROBE checklist submitted, this is a prerequisite for publication and all items should be reported in the manuscript
Author Response
Comment 1: The number of study participants is low and impairs making strong conclusions. Were there any sample size calculations or power considerations prior to conducting the study? Would have been worthwhile to consider what it would be possible to conclude with this study size.
Response 1: Thank you for your suggestion. In our study we included all refugees and asylum seekers that met inclusion criteria. The low number of participants and the fact that we are analyzing a “population” led us to not conduct inferential statistics
We confirm that we studied a small group, however these were the total number of participants who met the inclusion criteria in the study. Nevertheless, we discuss the small number of participants as a limitation of the study in 4.4. Limitations of the study Line 420: “The first limitation is the small number of study participants. This restriction is due to subjects that are not sampled from the whole population, but from a criterion-determined subpopulation, including having been welcomed and integrated into the CATR, between 2015 and 2020. Nevertheless, we assume that we don't have enough data to draw conclusions about entire populations.”
Comment 2: All findings depend on what the study participants report on their risk factors 12 months earlier. The authors acknowledge that this implies a huge risk of information bias, but should further consider that a number of other biases may apply as well as confounding, and this should be discussed in much more detail, preferably including the use of DAG's.
Response 2: Thank you for your suggestion. In our study we decided not to include DAGs because this tool is a simultaneous representation of association and causality with control of all confounders. The study in question is a cross-sectional study where it is not possible to establish cause and effect relationships between a condition and its risk factors or causes. However, the study's limitations were addressed in the results, in which we included more information about biases and possible confounders (Lines 420-434).
Comment 3: There is no STROBE checklist submitted, this is a prerequisite for publication and all items should be reported in the manuscript.
Response 3: Thank you! We have addressed this issue. We also indicated in the body of the manuscript. Line 121 “We followed STROBE guidelines for reporting observational studies (S1 STROBE Checklist).”